# The relative contribution of close-proximity contacts, shared classroom exposure and indoor air quality to respiratory virus transmission in schools

Nicolas Banholzer [1,2,3,14], James Daniel Munday [4,5,6,7,14], Philipp Jent [2,8], Pascal Bittel[2,9], Lorenzo Dall'Amico [10], Lavinia Furrer[9], Charlyne Bürki [4,5], Tanja Stadler [4,5], Matthias Egger[11,12], Tina Hascher[2,13], Ciro Cattuto [10] & Lukas Fenner [1,2] ✉

Close-proximity interactions are considered a key risk factor for respiratory virus transmission, but their importance relative to shared space and air quality remains unclear. We conducted a six-week longitudinal study in a Swiss secondary school (67 students, aged 14–15). We detected 87 infections in saliva samples and recorded absences to identify plausible transmissions, excluding implausible ones through genomic analysis. Time in close proximity (within 1.5 metres) was measured using wearable sensors and air quality via $CO_2$ monitors. Students spent 21.2 minutes per day in close proximity (interquartile range 7.8–44.2) and 5.3 hours in shared classrooms (IQR 3.8–6.2), during which air quality was suboptimal for 1.9 hours (IQR 1.2–3.0). Using pairwise survival models, we found that transmission was more likely within than between classes. Close proximity was modestly associated with higher transmission risk overall (rate ratio 1.16 per doubling daily time, 95%-CI 1.01–1.33), while time in shared classrooms (RR 3.17, 95%-CI 1.96–5.17) and suboptimal air quality (RR 1.90 95%-CI 1.23–2.94) also predicted within-class risk. Prolonged exposure in shared, poorly ventilated spaces, which potentially includes several infectious sources, drives respiratory virus transmission more than close contact.

Infectious diseases transmitted through the air route such as SARS-CoV-2 and Influenza, cause high morbidity and mortality, and their spread is difficult to control[1]. Closed spaces, crowded places, and close-contact settings (the 'three C's') are known to contribute substantially to transmission[2]. These conditions frequently converge in educational settings, where students experience prolonged interactions in confined indoor spaces, often in close proximity, facilitating transmission of respiratory infections, both in the context of

[1]Institute of Social and Preventive Medicine, University of Bern, Bern, Switzerland. [2]Multidisciplinary Center for Infectious Diseases, University of Bern, Bern, Switzerland. [3]Department of Public Health, University of Copenhagen, Copenhagen, Denmark. [4]Department of Biosystems Science and Engineering, ETH Zürich, Zürich, Switzerland. [5]SIB Swiss Institute of Bioinformatics, Lausanne, Switzerland. [6]Department of Epidemiology and Public Health, Swiss Tropical and Public Health Institute, Allschwil, Switzerland. [7]University of Basel, Basel, Switzerland. [8]Department of Infectious Diseases, Inselspital Bern University Hospital, University of Bern, Bern, Switzerland. [9]Institute for Infectious Diseases, University of Bern, Bern, Switzerland. [10]ISI Foundation, Turin, Italy. [11]Population Health Sciences, Bristol Medical School, University of Bristol, Bristol, UK. [12]Department of Infectious Diseases and Hospital Epidemiology, University Hospital Zurich, University of Zurich, Zurich, Switzerland. [13]Institute of Educational Science, University of Bern, Bern, Switzerland. [14]These authors contributed equally: Nicolas Banholzer, James Daniel Munday. ✉e-mail: lukas.fenner@unibe.ch

pandemics and seasonal epidemics[3–6]. Schools also contribute considerably to the overall transmission in the population as children transmit infection on to household members and into their communities[3,7–9].

Several studies have assessed close-contact patterns in schools[10–12], yet without quantifying their effect on transmission, especially not in relation to other risk factors. Empirical studies are scarce but indicate that close-range interactions may be needed for transmission[13–18], although prolonged exposure at longer ranges may incur a similar risk[13]. Specifically, proximity was associated with infection with SARS-CoV-2 during a long-distance train ride[16] and *Mycobacterium tuberculosis* during an airplane flight[17]. While most studies relate to SARS-CoV-2[13,14,16,18], recent work used a viral challenge model to study the transmission of multiple respiratory viruses (influenza virus A virus, respiratory syncytial virus, adenovirus, etc) following 30 min close-contact interactions between infectious children and susceptible adults[15]. Evidence from the SARS-CoV-2 pandemic further suggests that indoor air quality plays a critical role in the transmission of respiratory infections[19]. However, the relative importance of different risk factors remains inadequately quantified, limiting our ability to design targeted interventions for effective public health recommendations.

Here, we conducted a six-week longitudinal study during the peak of the respiratory virus season in winter of 2023/24 in a Swiss school. The study comprised four classes from the same grade level, all located on the same building floor to allow for interactions within and between classes. We used a comprehensive probabilistic framework combining molecular data, epidemiological data and assumptions, and genetic proximity of respiratory viruses to reconstruct plausible transmissions between students (Fig. 1). Furthermore, we detected close-proximity interactions using wearable sensors, monitored indoor air quality with aerosol devices and $CO_2$ monitors, and considered social factors possibly associated with transmission outside school. Using pairwise accelerated failure time models, we estimated the association of respiratory virus transmission with time in close-range proximity, time spent in shared classrooms, time with suboptimal air quality ($CO_2$ levels or particulate matter mass concentrations above recommended levels), and social factors (household siblings and extracurricular activities). We evaluated the relative contribution of each risk factor both overall and within individual classes, identifying key drivers of respiratory virus transmission in schools.

## Results

### Data collection overview: molecular, epidemiological, proximity, and environmental data

We collected molecular (saliva samples, viral genomic sequences), epidemiological (school absences, student characteristics), environmental (indoor $CO_2$ levels, particulate matter mass concentrations), and physical proximity data (time-resolved pairwise proximity relations) in four classes (67 students aged 14–15 years) of a secondary school in the canton of Bern, Switzerland, for six weeks from January 22 to March 8, 2024 (the study period for class 4 ended one week earlier on March 1).

We collected 1047 saliva samples from 67 out of 84 (80%) participating students. In their saliva, we detected 87 respiratory virus infections: 19 (22%) influenza viruses (IAV/IBV), 28 (32%) human

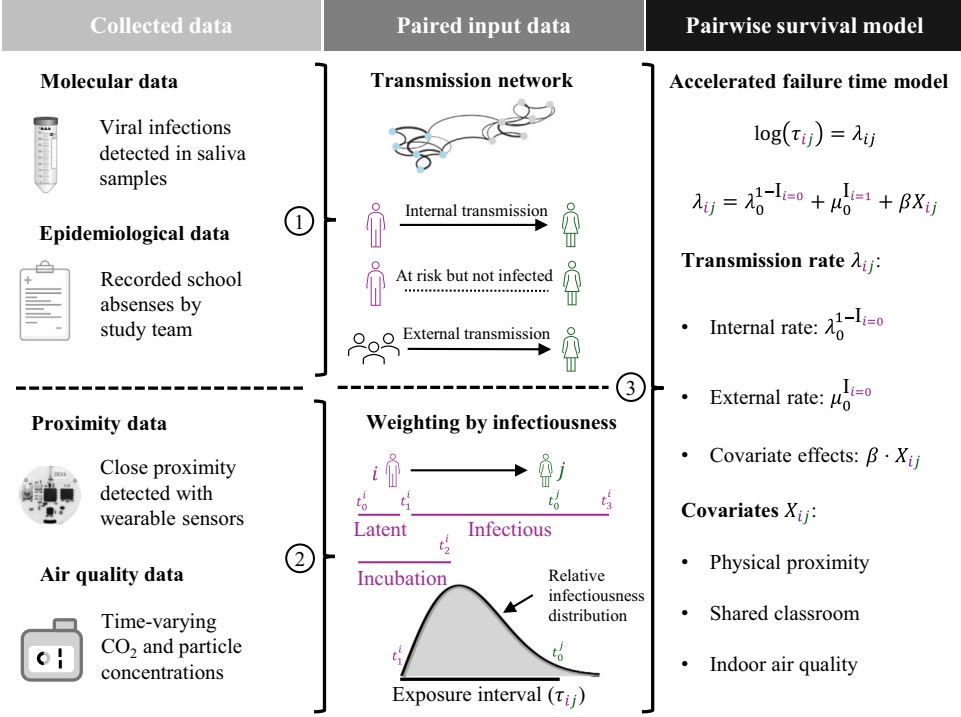

**Fig. 1 | Schematic overview of the collected and input data for the pairwise survival model.** (1) We detected virus infections in saliva samples and constructed paired datasets with transmissions that were epidemiologically plausible, excluding pairs where either the infectious or exposed student was absent from school. We also excluded transmissions of influenza A and respiratory syncytial virus through genomic analysis. If an infection could not be linked internally to another participating student, it was considered a transmission from an external source such as a household or community member. (2) For each student pair, we determined the time in close proximity during the exposure period, which was collected with wearable sensors worn by students throughout school lessons and break times. For student pairs within classes, we determined time spent in shared classrooms and in suboptimal air quality, which was measured with air quality monitors and aerosol devices. Both covariates were weighted by relative infectiousness of the infector and averaged over the exposure interval, which is the period from the onset of infectiousness of student $i$ (purple) until the date of infection of exposed and susceptible student $j$ (green). If $j$ is never infected, the exposure period is until the end of the infectious period of $i$. School-free days and absences are not included in the exposure period. (3) The association of respiratory virus transmission with time in close proximity and other risk factors was estimated using accelerated failure time regression models, with an internal and external rate of transmission.

**Table 1 | Study population and summary of viral infections and school absences overall and by school class**

| Variable N (%) Mean (SD) | Overall | K | N | P | S |
|---|---|---|---|---|---|
| **Participants** | **67/84 (79.8)** | **17/24 (70.8)** | **18/18 (100)** | **15/18 (83.3)** | **17/24 (70.8)** |
| Age range | 14–15 | 14–15 | 14–15 | 14–15 | 14–15 |
| Female | 26 (38.8) | 5 (29.4) | 6 (33.3) | 6 (40.0) | 9 (52.9) |
| Male | 41 (61.2) | 12 (70.6) | 12 (66.7) | 9 (60.0) | 8 (47.1) |
| **Viral infections** | **87** | **25** | **14** | **18** | **30** |
| IAV | 13 (14.9) | 8 (32.0) | 4 (28.6) | 0 (0.0) | 1 (3.3) |
| IAB | 6 (6.9) | 4 (16.0) | 2 (14.3) | 0 (0.0) | 0 (0.0) |
| HRV | 28 (32.2) | 2 (8.0) | 3 (21.4) | 7 (38.9) | 16 (53.3) |
| AdV | 10 (11.5) | 4 (16.0) | 0 (0.0) | 3 (16.7) | 3 (10.0) |
| PIV | 5 (5.7) | 0 (0.0) | 2 (14.3) | 1 (5.6) | 2 (6.7) |
| RSV | 11 (12.6) | 3 (12.0) | 2 (14.3) | 5 (27.8) | 1 (3.3) |
| HCoV-OC43 | 8 (9.2) | 2 (8.0) | 1 (7.1) | 2 (11.1) | 3 (10.0) |
| HCoV-229E | 6 (6.9) | 2 (8.0) | 0 (0.0) | 0 (0.0) | 4 (13.3) |
| **School absences** | **105** | **28** | **29** | **27** | **21** |
| Respiratory illness | 81 (77.1) | 23 (82.1) | 22 (75.9) | 22 (81.5) | 14 (66.7) |
| Other illness | 4 (3.8) | 1 (3.6) | 1 (3.4) | 1 (3.7) | 1 (4.7) |
| Unrelated to illness | 20 (19.1) | 4 (14.3) | 6 (20.7) | 4 (14.8) | 6 (28.6) |

*IAV* influenza A, *IBV* influenza B, *HRV* human rhinovirus, *AdV* adenovirus, *PIV* human parainfluenza virus, *RSV* respiratory syncytial virus, *HCoV-OC43* human coronavirus OC43, *HCoV-229E* human coronavirus 229E.

rhinoviruses, 10 (11.2%) adenoviruses, 5 (6%) parainfluenza viruses, 11 (13%) respiratory syncytial viruses (RSV A/B), and 14 (16%) human coronaviruses HCoV-229E and HCoV-OC43 (Table 1). We further recorded 105 absences from school, most of them related to an illness with at least one respiratory symptom.

After post-processing the wearable sensor data, we obtained close-proximity interactions (within 1.5 metres) between each student pair with a temporal resolution of 10 s. Overall, students were in close proximity with another student for a median of 21.2 min per day (IQR 7.8–44.2), mainly with students of their class (19.7 min, IQR 7.0–42.0). The proximity networks shown in Fig. 2 for classes N and P appear denser (5.2% and 3.6% of all possible edges) than those for classes S and K (1.3% and 2.7%). The daily cumulative time in close proximity for student pairs with at least one interaction was 0.7 min (IQR 0.3–3.0).

We measured $CO_2$ levels in all classrooms throughout the day using $CO_2$ aerosol devices and air quality monitors. Air quality was deemed suboptimal if $CO_2$ levels exceeded 1000 ppm. Students of the same class shared a classroom for 5.3 h per day (IQR 3.8–6.2), during which air quality was suboptimal for 1.9 h (IQR 1.2–3.0). Over the study (Fig. 3), $CO_2$ levels were above 1000 ppm for 43% of the time, with a lower proportion in classrooms N and P (27% and 38%) than in classrooms K and S (61% and 43%).

**Genomic analysis**

We sequenced the viral RNA from samples that tested positive for Influenza A, Influenza B and Respiratory Syncytial Virus (RSV) using an illumina targeted next-generation-sequencing workflow. Following alignment, consensus sequences were established for 10/13 IAV and 8/11 RSV A/B samples with positions masked with read-depths of <10. Coverage varied from 16–93% for IAV and 10–86% for RSV (Supplementary Fig. S1). Due to their low number and relatively poor quality, IBV sequences were not analysed further. We found 0–0.025 substitutions per comparable pair of bases for IAV and 0–0.42 for RSV. Using pairwise comparison of the consensus sequences, based on the

likelihood of finding pairs of sequences in subsequent hosts, we were able to identify 12 plausible transmission pairs for IAV and 6 for RSV (Supplementary Figs. S3 and S6). Repeating the analysis with consensus sequences masked at read depths of <5 and <20 had minimal impact on the results (Supplementary Figs. S2, S4, S5 and S7).

Figure 4 shows maximum likelihood phylogenetic trees generated using a coalescent model via the *Nextstrain* framework[20]; we present sequences acquired through this study alongside global sequences, accessed through GenBank, and sequences from the broader Swiss population, acquired in a parallel study[21]. IAV sequences were clustered together on the global tree, suggesting they were closely related compared to global diversity. Although the cluster was placed in a clade with other sequences from Switzerland, the cluster's location was uncertain due to genetic divergence from the rest of the tree. Genetic divergences within the school cluster were also larger than expected within the study's timeframe, suggesting they were part of an outbreak in the local community or school. Differences in the parts of the genome that were sequenced successfully resulted in non-transitive comparison between sequences. As a result, some identical sequences were placed together (e.g., K-75, K-32 and K-95), but others were not (e.g., K-32 and K-95 relative to S-7). This highlights that, while our analysis was able to rule out transmissions, not all remaining pairs were jointly feasible.

Four RSV G sequences were identical in comparable sections, of which three were clustered together on the tree. The fourth was within the same clade but not adjacent to the other three. This suggests that there were at least three introductions of RSV into the school classes. Considering that most of the positive RSV samples were collected within the first few days of the study, it is likely that observed infections form the tail of an outbreak before our observations began (Supplementary Fig. S8).

**Transmission networks**

We used the pairwise survival analysis framework to determine epidemiologically plausible transmissions between students based on assumptions about the pathogen-specific incubation and infectious period distribution. Transmissions were excluded if either student was completely absent from school during the infectious period, the exposed student was considered immune due to a prior infection, or the transmission was excluded through analysis of the genomic sequences.

Considering uncertainty in the epidemiological parameters, the probabilistically generated datasets comprised 4961 (IQR 4905–5021) possible pairs where a susceptible student was exposed to infection. Across pathogens, a median of 84 (IQR 75–92) transmissions between students and 38 (IQR 35–40) with a probable external source (e.g., a non-participating student, a household or a community member) were identified. The transmission networks of influenza A and respiratory syncytial virus with genomic exclusions are shown in Fig. 5 and the networks for all other respiratory viruses are shown in Supplementary Fig. S9–S12, including the networks for influenza A and respiratory syncytial virus without genomic exclusions. Most IAV and RSV transmissions occurred within classes. Note that IAV infections were close in time, allowing circular transmission chains to be included in the resulting network.

**Association of transmission with risk factors**

We used pairwise accelerated failure time regression models to estimate the association between respiratory virus transmission and close proximity, shared classroom time, air quality, and social factors. Time to event was the number of days until infection of the susceptible student, excluding days when the susceptible student was not exposed (absent from school) while the other student was infectious. Prior to estimation, time in close proximity, shared classrooms, and suboptimal air quality were weighted by relative infectiousness to give

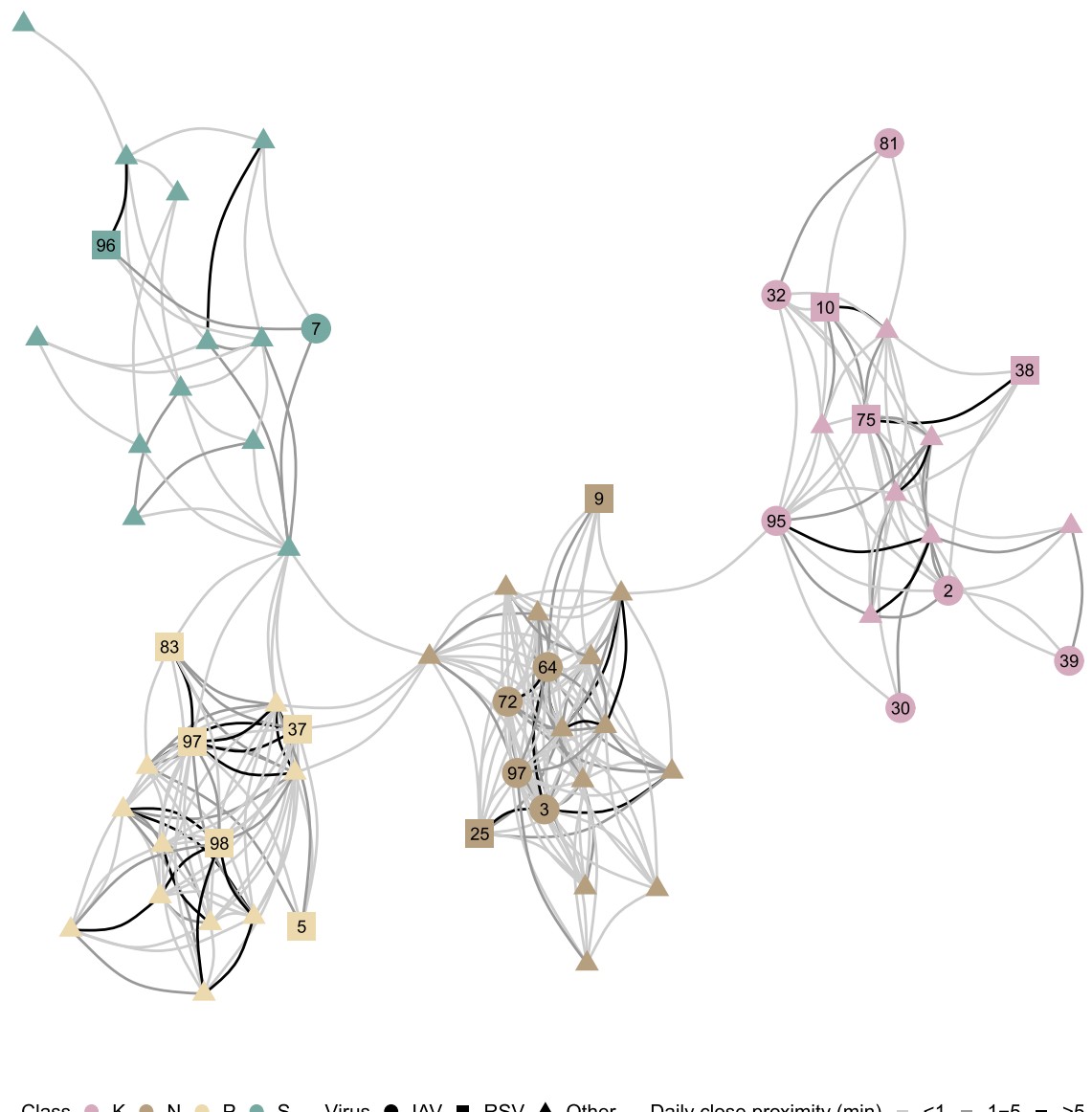

**Fig. 2 | Close proximity data.** Contact network showing the median daily time spent in close proximity. Connections are displayed for student pairs who interacted at close range on at least half of the days they attended school, with thicker lines indicating longer time in proximity. Nodes are coloured by school class.

Student IDs are shown for individuals infected with influenza A virus (IAV, circles) or respiratory syncytial virus (RSV, squares), while triangles indicate students infected with other viruses. Corresponding transmission networks for IAV and RSV are shown in Fig. 5.

higher weight to exposures around symptom onset of the infectious student.

Based on the model estimates aggregated across all probabilistically generated datasets, a twofold increase in the infectiousness-weighted daily time in close proximity was associated with a 16% higher rate of transmission (rate ratio [RR] 1.16, 95%-credible interval [CI] 1.01–1.33). After five days of exposure, the transmission risk was about 0.5% without any close-range interaction, about 1.5% with one minute, and about 2.5% with ten minutes in close proximity per day (Fig. 6A). The median time in close proximity during the study and the unweighted average time during the exposure period were only tentatively associated with transmission (Supplementary Table S1). Results were similar when using physical proximity data with different attenuation thresholds and transmission data without genomic exclusions (Supplementary Fig. S13).

Transmission was more likely within than between classes (RR 4.02, 95%-CI 1.69–9.64). After five days of exposure, the transmission

risk was about 0.5% between versus about 2% within classes (Fig. 6B). A twofold increase in shared classroom time was associated with higher transmission (RR 3.17, 95%-CI 1.96–5.17). We did not find evidence for an association of time in close proximity with transmission within classes (RR 1.03, 95%-CI 0.88–1.21; see Supplementary Fig. S13 for different attenuation thresholds).

A twofold increase in the time $CO_2$ levels exceeded 1000 ppm (suboptimal air quality) was associated with higher transmission (RR 1.90, 95%-CI 1.23–2.94). After 5 days of exposure, the transmission risk remained below 1–2% for 1–2 h per day with suboptimal air quality and reached almost 4% for four hours (Fig. 6C). Associations were similar when using slightly lower or higher $CO_2$ level thresholds for suboptimal air quality, or when measuring it with the time during which the concentration of particles smaller than 2.5 μm ($PM_{2.5}$) exceeded 5 μm/m³ (Supplementary Table S1).

In addition to estimating relative risks, we evaluated the relative contribution of each risk factor to transmission based on the relative

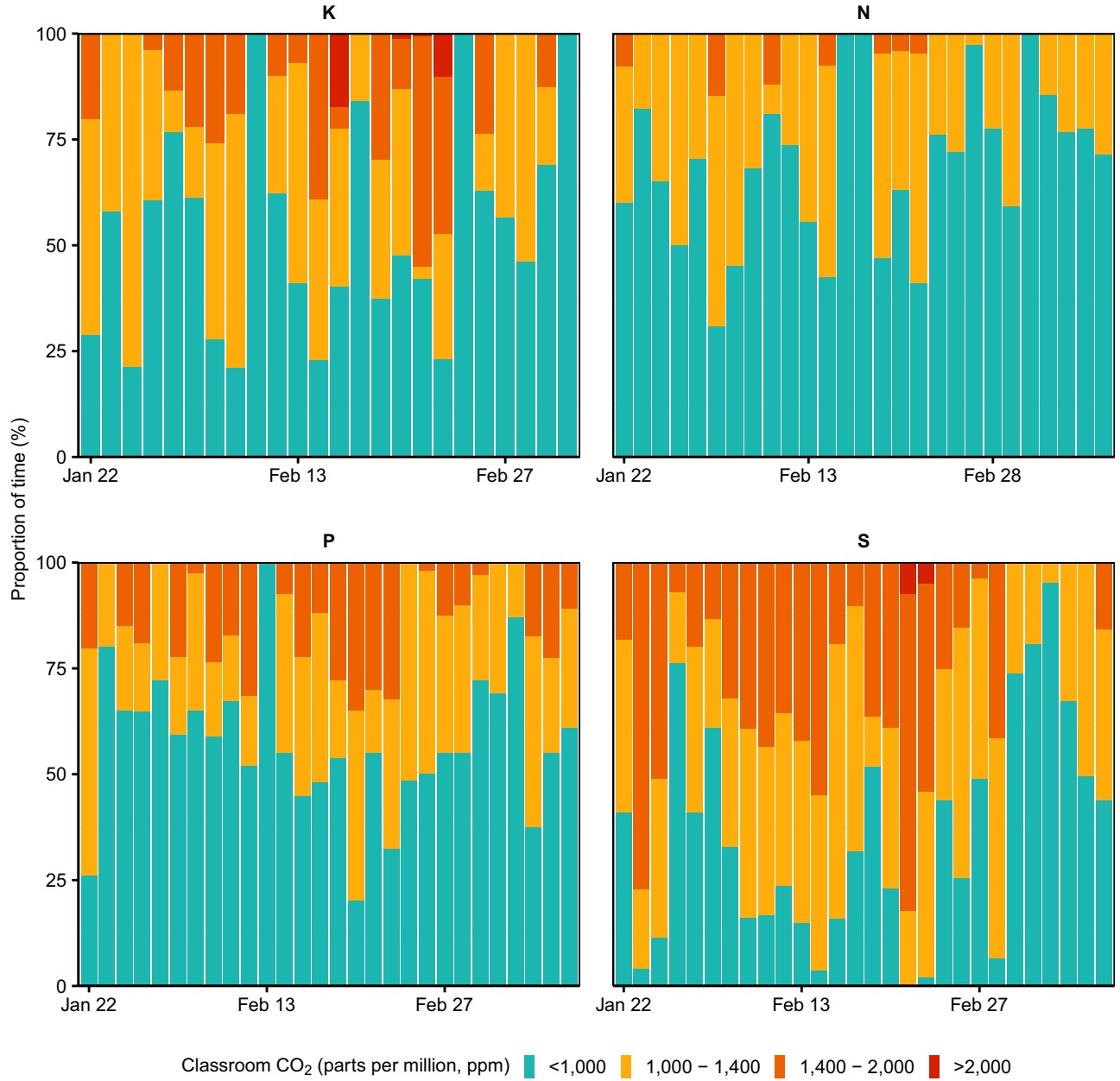

**Fig. 3 | Air quality data.** Indoor air quality by study day and school class measured as the proportion of time (%) classroom $CO_2$ levels exceeded 1000, 1400, and 2000 parts per million (ppm), respectively. Stacked bars indicate these proportions and are coloured by $CO_2$ threshold.

changes in the model log-likelihoods over the null model without any covariates. The relative contributions are shown in Fig. 6D. Prolonged exposure in shared classrooms (relative contribution 54%, 95%-CI 22–81) tended to contribute more to overall transmission than time in close proximity (21%, 95%-CI 0–44). For transmissions within classes, shared classroom time (relative contribution 59%, 95%-CI 22–80) and suboptimal air quality (29%, 95%-CI 5–49) contributed more than time in close proximity (4%, 95%-CI 0–20). The relative contributions of social factors (household siblings and extracurricular activities) were about 10% overall and about 5% for transmissions within classes. In terms of relative risks, there was little evidence for an association of social factors with transmission (Supplementary Table S1).

## Discussion

We studied the association of respiratory viral transmission in a Swiss school with time in close proximity to an infectious student, prolonged exposure in shared classrooms, and indoor time with suboptimal air

quality. We used a multifaceted approach combining molecular, physical proximity and environmental data within a probabilistic framework. We identified probable transmission networks that were suggestive of transmissions in school, mostly within but sometimes also between classes. Our study provides further empirical evidence that close-range interactions contribute to respiratory virus transmission. However, persistent exposure to contaminated shared classroom air was the dominant risk factor. This comprehensive assessment in a real-world educational setting offers new insights into the transmission dynamics among adolescent students based on objective measures of physical proximity, virus infections and environmental conditions.

We found that time spent in proximity was associated with a higher risk of transmission. Previous studies in confined environments have also observed that susceptible individuals were more likely to be infected when in closer proximity to infectious individuals[16,17,22–24]. In a virus challenge model in a healthcare setting, close-range contact for

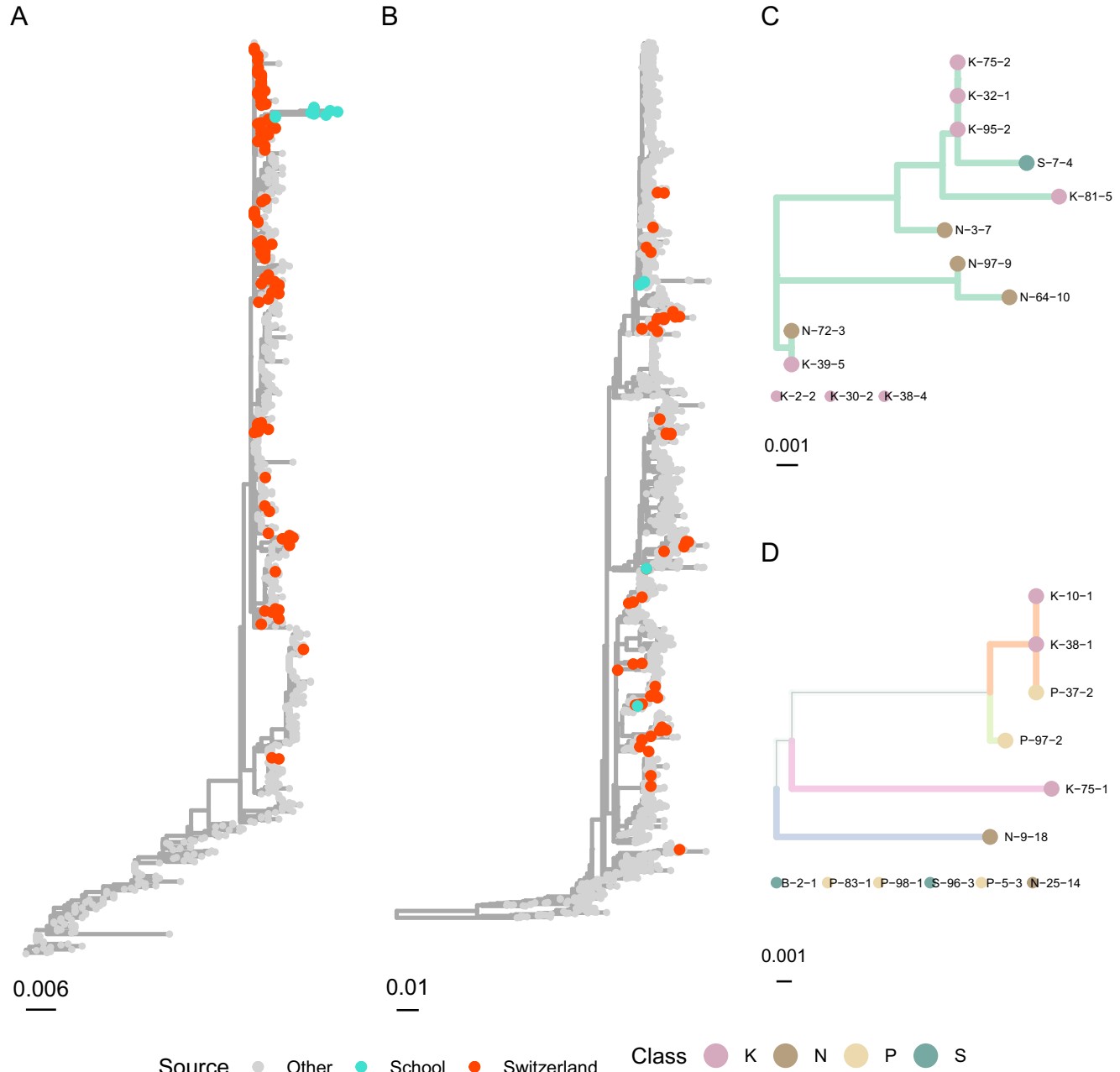

**Fig. 4 | Maximum likelihood phylogenetic mutation trees for influenza A and respiratory syncytial virus. A** Tree for influenza A H1N1 NA gene and (**B**) respiratory syncytial virus G gene, including international samples. International samples are coloured in grey, Swiss samples in red, and samples from this study in green. Also phylogenetic mutation trees with all non-ancestral branches for the study samples pruned for (**C**) Influenza A H1N1 NA gene and (**D**) respiratory

syncytial virus G gene. Branch colours indicate sections of the tree that are entirely occupied by samples from this study. Branches are shown in a common colour if they represent a complete sub-tree on the global tree, hence branches of different colours indicate that non-school-study samples exist between the constituent leaves. Nodes are coloured by school class.

30 min resulted in transmission in 15% of interactions between infectious children and adults[15], supporting the plausibility of our real-world findings in schools. Furthermore, a digital contact tracing study of SARS-CoV-2 transmission found that short close-range exposures incurred a similar risk as prolonged long-range exposures[13]. In our study, time in close proximity was associated with transmission when weighted by relative infectiousness, suggesting that close contact around symptom onset facilitates transmission. Within classrooms, time in close-range proximity was not clearly associated with transmission. Instead, duration of exposure and air quality predicted transmission events. This suggests that the cumulative risk from shared presence in a contaminated indoor space may exceed the risk

of short-range person-to-person transmission, possibly because every susceptible individual is generally only exposed to a small number of potentially infectious sources via close contact compared to multiple ones in a classroom.

Indoor air quality emerged as a critical risk factor for transmission within classrooms, with more transmissions in school classes with lower air quality throughout the study. Most infections occurred in the school class where $CO_2$ exceeded 1000 ppm during two-thirds of the time students were in the classroom compared to one-third of the time in the school class where only few infections occurred. Previous work has shown that poorly ventilated indoor spaces facilitate the transmission of respiratory infections[6,25,26], also

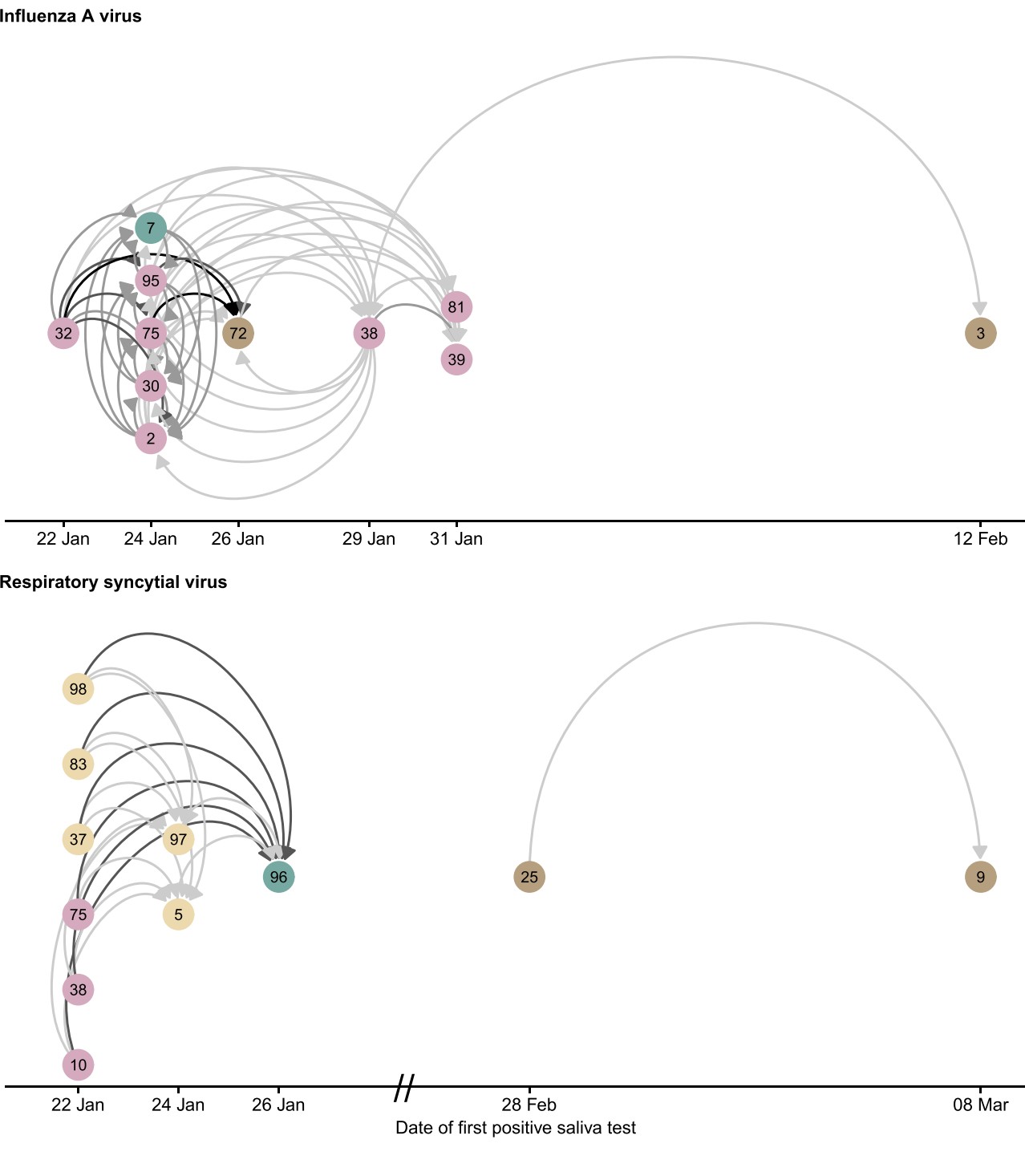

**Fig. 5 | Transmission networks of influenza A and respiratory syncytial virus based on epidemiological and genomic data.** The networks show all plausible within-school transmission events between participating students, based on epidemiological delays and school absences, with implausible links excluded through genomic analysis. Arrow thickness indicates the probability of transmission, defined as the proportion of paired datasets in which each link was present, accounting for uncertainty in the timing of infections and infectious periods. The temporal axis represents the date of the first positive saliva sample, which may occur several days after infection, allowing circular links. Nodes are coloured by school class.

over longer ranges depending on airflow[27], and including the strictly airborne bacterial pathogen *Mycobacterium tuberculosis*[28,29]. However, most studies are modelling-based[30–33] and lack the combination of multiple data sources to first determine transmission and then its association with environmental and other risk factors. Our

results could be applied in mathematical models to refine estimates about the impact of infection control and preventive measures in schools and other indoor settings[10,34]. They also emphasise the importance of indoor air quality in building designs and ventilation performance[19,26,35].

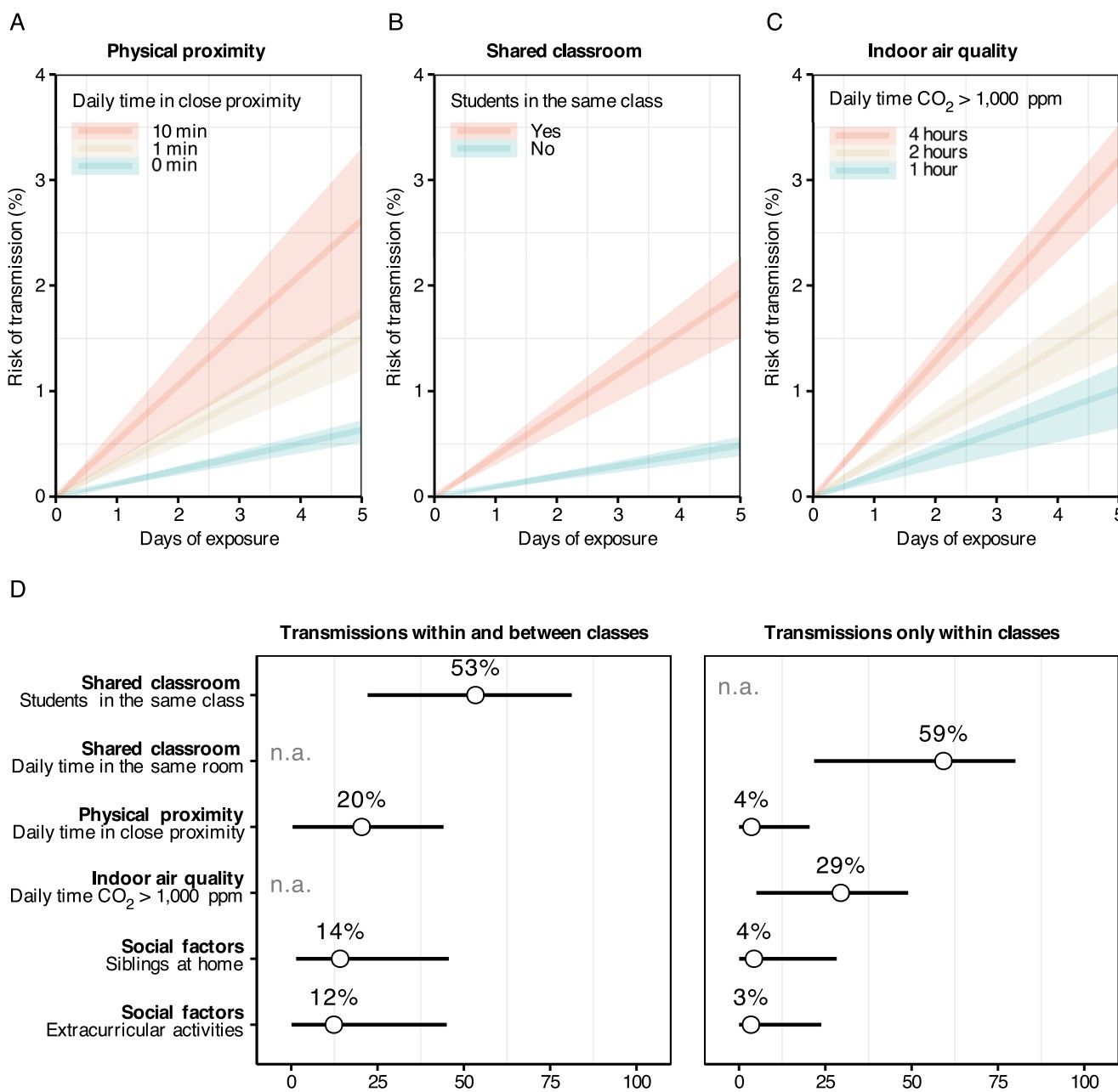

**Fig. 6 | Association of respiratory virus transmission with close proximity, shared classroom environment and indoor air quality. A–C** Estimated transmission risk by days of exposure over one school week, shown as mean estimates (lines) with 95% confidence intervals (ribbons) coloured by different covariate levels. **A** Transmission risk by daily time spent in close proximity for both within- and between-class transmissions. **B** Transmission risk for students depending on whether they shared a classroom. **C** Transmission risk by daily duration that classroom $CO_2$ levels exceeded 1000 ppm (indicative of suboptimal air quality), restricted to within-class transmissions. **D** Relative contribution of each risk factor to transmission, expressed as the change in model log-likelihood when adding each predictor separately to a baseline (null) pairwise accelerated failure time model. Relative factor contributions are shown with means (dots) and 95% quantiles (lines) across the $N = 1000$ probabilistically generated datasets, representing uncertainty due to the stochastic reconstruction of transmission pairs. Note that the contributions of shared classroom time and low air quality could not be assessed (n.a.) for all transmissions combined (within and between classes), and the effect of being in the same class could not be assessed for within-class transmissions alone.

Our findings should be interpreted in the context of the different routes and particle sizes for the transmission of respiratory viruses via the air[36–38]. Short-range transmission occurs when respiratory particles of various sizes are transmitted over short distances through direct projection or inhalation, requiring proximity between infectious source and susceptible host. In contrast, long-range transmission involves smaller particles that remain suspended in the air, so that they can accumulate in poorly ventilated rooms, increasing the risk of infection for everyone in the room, irrespective of proximity. Understanding transmission within the framework of the "three C's"—closed spaces, crowded places, and close-contact settings—has therefore emerged as a critical paradigm for respiratory infection control[2]. While these conditions frequently coincide in educational settings, their relative importance remains poorly understood. In addition, the relative contributions of short and long-range transmissions to overall transmission may vary by pathogen. For example, a comparison of our

previous studies in schools showed that SARS-CoV-2 was more frequently detected in air samples than other respiratory viruses[39], suggesting greater persistence in the air and a potentially prolonged exposure risk in shared indoor environments.

Our genomic analysis revealed distinct transmission clusters within classrooms, with limited cross-class transmission despite students sharing the same building floor. This emphasises how the classroom microenvironment creates conditions favouring virus spread beyond what contact patterns alone would predict. While previous studies have sought to determine transmission dynamics in close-proximity environments[40,41], our study integrates molecular epidemiology, genomic analysis, and proximity data from wearable sensors within a probabilistic framework to reconstruct transmission networks in schools.

Our study has several limitations. First, our six-week study represents a single season in a Swiss school. Several influenza A and respiratory syncytial virus infections were observed at the start of the study. This pattern, combined with our genomic analysis, suggests that we may have missed the peak of transmission for these viruses and that some students were no longer susceptible to infection. Transmission dynamics may also be different in other seasons or regions, and vary depending on building designs or educational characteristics influencing contact behaviour and indoor exposure. Nevertheless, the study was overall conducted during the peak of the respiratory virus season and our comprehensive viral test panel captured multiple circulating pathogens, providing broader insights than single pathogen studies. Second, incomplete genomic sequence coverage for pathogens other than influenza A and respiratory syncytial virus limited the inference of transmission networks. Our probabilistic framework considered a wide range of scenarios, including school absences and uncertainty about the timeline of infections. However, it may still underestimate the number of external transmissions, as any transmission will be categorised as internal if the timeline of the person who became infected (infectee) aligns with that of at least one plausible source of infection (infector) among the participating students. Third, although more participants across multiple schools would strengthen our results, our data-intensive longitudinal design with high-frequency sampling provides an in-depth analysis that compensates for breadth. We note, however, that the sample size per pathogen was not large enough to study variation in the effects of transmission risk factors between pathogens. Fourth, although $CO_2$ concentration is an established proxy for ventilation efficiency, it does not directly reflect viral concentration in the air. Fifth, wearable sensors measure time in proximity but not the type of activity, which might modify transmission risk. Finally, we note that the estimated relative contributions might be partially driven by the fact that exposure in the classroom is usually to several potential infectious sources, whereas close proximity is limited to a single or few potential sources at any given time.

In conclusion, our study shows that, while time in close proximity increases the risk of respiratory virus transmission, prolonged exposure to contaminated air in confined spaces overall emerges as the dominant driver of viral spread in indoor environments such as educational settings. Our results suggest a complex interplay between prolonged exposure, crowding and environmental risk factors, as well as a relevant contribution of the number of potential infectious sources present in a room. Indoor air quality was associated with intraclass transmission, highlighting how inadequate ventilation creates ideal conditions for virus transmission. Our findings also have important implications for public settings other than schools, such as workplaces, transportation, and healthcare facilities. Importantly, they support the adoption of comprehensive infection control strategies that address both crowding and environmental interventions designed to improve indoor air quality. These strategies will inform the development and evaluation of interventions in the context of building design, seasonal epidemics, and pandemic preparedness. Future

studies across diverse settings, possibly with real-time monitoring of air quality, could enable adaptive risk assessment and targeted interventions. Our multifaceted approach offers a promising direction for holistic environmental and digital surveillance of infectious disease transmission.

## Methods

### Study design and setting

We collected molecular (saliva samples, viral genomic sequences), epidemiological (school absences, student characteristics), environmental (indoor $CO_2$ levels, particulate matter mass concentrations), and physical proximity data (time-resolved pairwise proximity relations) in four classes (67 students aged 14–15 years) of a secondary school in the canton of Bern, Switzerland, for six weeks from January 22 to March 8, 2024 (the study period for class 4 ended one week earlier on March 1).

The classes were located on the same building floor (Fig. 7), allowing students of different classes to interact during breaks and personal study. On a typical weekday, the students arrived at the school around 7:30 am and lessons ran until noon, with small 5–10 min breaks between lessons and a longer 30 min break at 10 am, during which they wore the sensors. After a 1-h lunch break, often outside school and without wearing the sensors, lessons continued in the afternoon until about 3:30 pm. Most indoor lessons took place in shared classrooms among students of the same class, but students from different classes could interact during short breaks in the corridor, long breaks on the school ground, and during times reserved for personal study in shared rooms.

At study start, students were asked to report the number of siblings in their household and any extracurricular activities, which were considered social factors of transmission outside school. School absences and times spent in classrooms were recorded daily by our local study team and entered electronically on the secure web platform REDCap (https://project-redcap.org/)[42].

### Molecular data

Saliva samples were collected three times per week (Monday, Wednesday, Friday) and analysed by real-time PCR using the Seegene's Allplex RV Master Assay and the Respiratory Panel 3 (Seegene, Seoul, South Korea) to detect a combined panel of 24 major respiratory viruses and viral subtypes, including SARS-CoV-2, influenza A/B virus, respiratory syncytial virus, adenovirus, metapneumovirus, parainfluenza virus, rhinovirus, coronaviruses NL63, 229E, OC43, and bocavirus. We defined an infection episode as a consecutive period of positive saliva test results and separated them into two episodes if more than one week elapsed between two positive results. Our trained study team further collected daily data on absences from school. We refer the reader to ref. 43 for a detailed description and analysis of the molecular and epidemiological data of this study.

### Physical proximity data

We used wearable sensors from the SocioPatterns collaboration (sociopatterns.org) to detect close proximity between students. The methodology has been used in many settings[11,40,41,44–47], including schools[11,44], households[41,45], social gatherings[47], tertiary care settings[40], and has been validated against self-reported contacts recorded via paper diaries[46]. The wearable sensors detect close-range proximity by exchanging ultra-low-power radio packets and measuring attenuation and reception rate. The data logged by individual sensors can be post-processed applying application-specific definitions of "contact" based on signal attenuation, reception rate, and more. For the present study, data were post-processed with an attenuation threshold of −65 dBM so that close-range interactions within about 1–1.5 m were detected. This choice was made before analysis began, considering the radio setup and enclosures used for data collection, and following the choices

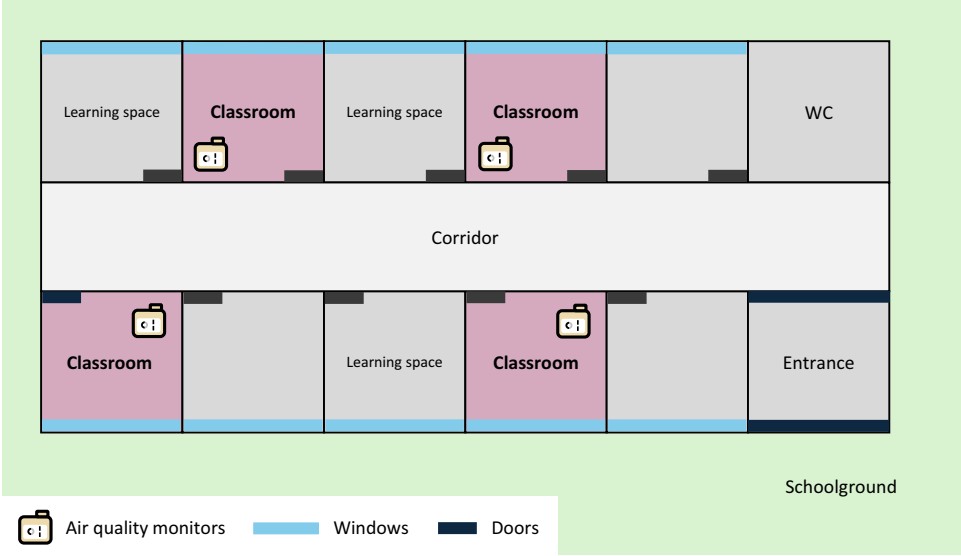

**Fig. 7 | Schematic view of the school study setup.** The four classrooms (purple squares) were on the same floor of the building, separated by a corridor. There were also rooms for personal study and group work, where students from different classes could meet. During breaks, students could choose to spend their time indoors or outside on the schoolground. Each room has a door onto the corridor and a window overlooking the surrounding schoolground. One air quality monitor was installed per classroom.

made in previous studies with similar setups and settings[41,48]. To assess variation, we performed a sensitivity analysis with a slightly higher and lower threshold compared to our default, which detects more and less close-proximity encounters at larger and shorter distances, respectively.

Close-proximity contacts were measured with a temporal resolution of 10 s, yielding a time-resolved proximity graph between study participants. Each sensor has a unique ID linking the contact data to its wearer. Students wore the sensors pinned to the front of their shirts in a pouch to ensure consistent geometry and more accurate proximity detection. They were asked to wear the sensors throughout school hours, except during sports and school trips. Data was downloaded from the sensors at the end of every second week, when sensors were serviced to replace batteries and clear on-board memory. Wearable sensors also included a 3D accelerometer that records the device's orientation in space and over time, which was used to detect compliance with sensor wearing. Moreover, we addressed adherence issues (lost or replaced devices) and removed spurious interactions when sensors were collected or handed out. After post-processing, we computed the daily cumulative time in close proximity between each pair of students. If two students were in school but one of their sensors was not worn (as detected by the accelerometer, i.e., stationary for an extended time interval), the time was imputed with the daily median over the study.

### Air quality data

We collected $CO_2$ levels per classroom using two aerosol devices (AQ Guard, Palas GmbH, Karlsruhe, Germany) and two air quality monitors (Aranet4 Home, SAF Tehnika JSC, Riga, Latvia). We measured the time during which $CO_2$ levels exceeded 1000, 1400 or 2000 ppm, commonly used thresholds for suboptimal indoor air quality[49]. In two classes, we also measured the time during which the concentration of particles smaller than 2.5 μm ($PM_{2.5}$) exceeded 5 μg/m³, a threshold recommended by the Swiss Federal Commission on Air Hygiene[50].

### Genomic analysis

Viral genomes from samples testing positive for influenza (IAV/IBV) and respiratory syncytial virus (RSV A/B) were sequenced on NovaSeq (Illumina, USA) using an amplicon based next-generation-sequencing approach (xGen Respiratory Virus kit, IDT, USA). Reads were assembled using BLAST and consensus sequences established for samples with ≥20% coverage. Sites with read depth <10 were masked. Consensus sequences of IAV and RSV A were analysed to identify transmissions. Due to limited coverage, we focused on the NA region of IAV and the G gene of the RSV A genome.

To establish likely transmission events, we calculated the number of single nucleotide polymorphisms (SNPs) between each pair of sequences as the proportion of comparable sites. We then computed the likelihood that sequences were consistent with a transmission using the Jukes Cantor '69 substitution model. We assumed an evolution time of the number of days between test dates plus a perceived maximum of 4 days (2 days of within-host evolution per sample), and calculated the log-likelihood of the model using the closed form expression[51]

$$\log \mathcal{L}\left(\mu, t | s_i, s_j\right) = l\left(s_i, s_j\right) \log\left(\frac{1}{4} + \frac{1}{3}\exp\left(-\frac{4}{3}\mu t\right)\right) \\ + d\left(s_i, s_j\right) \log\left(\frac{1}{4} - \frac{1}{4}\exp\left(-\frac{4}{3}\mu t\right)\right) \quad (1)$$

where μ is the substitution rate, $t$ is the time between samples in evolutionary time, $s_i$ and $s_j$ are the sequences of samples collected from individuals $i$ and $j$, $d(s_i, s_j)$ and $l(s_i, s_j)$ are the hamming distance and the number of comparable nucleotide positions between $s_i$ and $s_j$, respectively. For both IAV and RSV, we assumed a nucleotide substitution rate of $1.5 \times 10^{-3}$ mutations per site per year, which is close to the estimates in literature[52,53].

We compared our sequences with those from clinical samples across Switzerland[21] and those publicly available from GenBank, constructing maximum likelihood phylogenetic trees using a coalescent model in the *Nextstrain* framework[20]. See Supplementary Text A for detailed methods.

### Pairwise survival analysis

To assess the association of virus transmission with close-range proximity and air quality, we used the pairwise survival analysis

**Table 2 | Assumptions about the pathogen-specific incubation and infectious period distribution**

| Pathogen | Incubation period distr. (median, dispersion) | Infectious period distr. (median and dispersion) | Sources |
|---|---|---|---|
| Influenza A | Lognormal (1.4, 1.51) | Lognormal (5, 1.19) | 57,58,61,64 |
| Influenza B | Lognormal (0.6, 1.51) | Lognormal (4, 1.16) | 57,58,61,64 |
| Adenovirus | Lognormal (5.6, 1.26) | Lognormal (4, 1.80) | 57,65 |
| Parainfluenza virus | Lognormal (2.6, 1.35) | Lognormal (7, 2.15) | 57,61,66 |
| Respiratory syncytial virus | Lognormal (4.4, 1.24) | Lognormal (4, 1.64) | 57,60,61,67 |
| Human rhinovirus | Lognormal (1.9, 1.68) | Lognormal (11, 1.17) | 57,68 |
| HCoV-OC43 | Lognormal (1.4, 1.51) | Lognormal (5, 1.19) | Same as for influenza A |
| HCOV-229E | Lognormal (1.4, 1.51) | Lognormal (5, 1.19) | Same as for influenza A |

framework[54–56]. Figure 1 provides a schematic overview from data collection to modelling. In summary, we identified virus infections from our molecular data using the first positive saliva test results. We probabilistically created datasets of all possible infector-infectee pairs among participating students, considering uncertainty in the pathogen-specific incubation and infectious period. In each dataset, we distinguished between internal transmission pairs among study participants and external transmission pairs with unknown external sources, as well as the corresponding internal and external pairs at risk with the susceptible students exposed but not infected. Times in close proximity and other risk factors were computed for each internal pair and the associations were estimated using accelerated failure time models, with time to event defined as the number of days the susceptible student was exposed while being in school together with the infectious student. The models were estimated for each paired dataset and the estimation results were summarised with the median across datasets. In the following, we provide a detailed description of the model input data, epidemiological assumptions, the approach to generate paired datasets, and model estimation. Note that we sometimes deviate from the original pairwise survival analysis framework[54–56] to tailor the analysis to our school setting. These deviations are noted accordingly. See Supplementary Text B for illustrations and examples of important methodological details.

### Timelines of infection from molecular data

An infection episode was defined as a period of consecutive positive molecular tests, starting from the first positive test result to the last one. Two episodes were separated if more than one week elapsed between two positive test results. Supplementary Fig. S8 shows an overview of the test results and school absences.

The date of the first positive saliva test result was assumed to be the start of infectiousness. We considered a possible delay in obtaining the test result due to days without testing (Tuesday and Thursday), school-free days (school holidays and weekends) and absences from school (due to illness or other reasons). For example, if a student first tested positive on Wednesday, we assumed that the first possible date was Tuesday. Similarly, if a student first tested positive on Friday but was not in school from Wednesday to Thursday, the first possible date was also Tuesday. We sampled the first possible positive test date probabilistically, assuming that dates near the first observed positive test results were more probable (as illustrated in Supplementary Fig. S14).

We backcalculated the date of infection (start of infection) from the first possible positive test date (corresponding to the start of infectiousness) by making assumptions about the pathogen-specific incubation and latent period. The end of the infectious period depended on the pathogen-specific infectious period. The end of the infection corresponded to the last possible positive test date, which was probabilistically sampled analogous to the first possible one. Immunity to reinfection with the same pathogen started from the date of infection and lasted until end of infection plus a pathogen-specific

immunity period. The timeline of infections is illustrated in Supplementary Fig. S15.

### Epidemiological assumptions

The incubation and infectious periods were sampled from pathogen-specific lognormal prior distributions (Table 2). The pathogen-specific incubation periods were obtained from a systematic review[57]. The latent periods were assumed to be one day shorter than the incubation period for influenza A/B, respiratory syncytial virus, parainfluenza virus[58–61], and human coronaviruses HCoV-229E and HCoV-OC43. They were assumed to be equal to the incubation period for the other respiratory viruses, implying no pre-symptomatic transmission.

The infectious period has rarely been systematically assessed for respiratory viruses. We used various sources, mostly pertaining to viral shedding, and expert knowledge to determine the infectious period per pathogen. If available, we used existing estimates, and otherwise formulated priors about the median, lower and upper limit, from which we computed the dispersion. Since we could not find estimates for the incubation and infectious period specific to human coronaviruses HCoV-229E and HCoV-OC43, we assumed their prior distributions were the same as for influenza A.

The immunity period was fixed at one week for human rhinovirus and eight weeks for all other respiratory viruses. Thereby, we exclude the possibility of re-infection over the study period for pathogens other than human rhinovirus. Re-infections were mainly observed for human rhinovirus and only in a few cases for adenovirus (Supplementary Fig. S8). For human rhinovirus, frequent re-infections are plausible in school settings due to the simultaneous circulation of numerous antigenically distinct serotypes[62], allowing individuals to be infected multiple times within a short period.

### Paired datasets of infectious and exposed students

We generated multiple datasets of epidemiologically plausible transmissions between pairs of students. For each infection, we formed pairs of the infectious student with all other students participating in the study. If either the infectious or the susceptible student was absent from school during the entire infectiousness period of the infectious student, we considered the susceptible student not exposed and removed the pair. We further assumed no exposure during school-free days. Finally, we removed pairs if the exposed student was still considered immune due to a previous infection, assuming no cross-immunity between pathogens, which means that, for example, a previous influenza A infection conferred no protection against a respiratory syncytial viral infection.

We first determined internal transmission pairs, defined as an infection of the exposed student with the same pathogen as the infectious student. For influenza A and respiratory syncytial virus, we excluded highly unlikely internal transmissions based on whether a pairwise comparison of genetic sequences yielded a likelihood of lower than 0.5 for transmission. Note that an exposed student can have multiple plausible infectors. Circular (bi-directional) transmissions are

also possible because the epidemiological parameters were sampled per individual, which means that in any probabilistically generated dataset two students testing positive for the same pathogen on the same day could have a different incubation period and thus a different infection date. These circles can further be amplified by variations in the first possible positive saliva test date due to absences and school-free days.

Infections that were not linked internally were assumed to have an external source outside the study. Pairs without any plausible transmission were denoted as pairs at risk, resulting in four types of pairs:

1. Internal transmission pair: Susceptible student infected by an internal source (participating student).
2. External transmission pair: Susceptible student infected by an external source.
3. Internal pair at risk: Susceptible student exposed to an internal source but not infected.
4. External pair at risk: Susceptible student exposed to an external source throughout the study and for every pathogen.

Our paired data represents time to event data, with time to event defined as the time to infectious contact, referred to as the contact interval[54–56]. However, in our school setting, susceptible students were not exposed internally, for example, when absent from school. Therefore, we defined the time to event for internal pairs as the number of days the susceptible student was exposed and in school together with the infectious student and refer to it as the exposure interval. Note that the exposure intervals can be right censored for two reasons. First, the start and end of the infectious period is censored by the start and end of the study period, respectively. Second, the exposure intervals for internal and external pairs at risk can be censored by prior internal or external infection. After considering censoring, a pair at risk was removed if it has an exposure interval of length zero. We provide examples of exposure intervals with and without censoring in Supplementary Fig. S16.

### Pairwise accelerated failure time models

The pairwise accelerated failure time regression model[54–56] is related to a susceptible, infectious, or removed (SIR) model. Assume individual $i$ is infected at time $t_0^i$, becomes infectious at $t_1^i$, develops symptoms at $t_2^i$, and ceases to be infectious at $t_3^i$. During the infectious period, $i$ can infect susceptible individual $j$ if they had contact while $i$ was infectious. The time until infectious contact is referred to as the contact interval $\tau_{ij}$, but we denoted it as the exposure interval (see above), also to distinguish it from (close) contact between $i$ and $j$, which is a covariate in our model.

Accelerated failure time regression models belong to the broader class of parametric survival models. They assume that covariates accelerate or decelerate the event, which in our analysis was respiratory virus transmission between individuals $i$ and $j$. The exposure interval $\tau_{ij}$ is drawn from a failure time distribution. We chose an exponential distribution with a cumulative hazard function

$$H_{ij}\left(\tau_{ij}\right) = \int_0^{\tau_{ij}} h(u)du = \tau_{ij} \cdot \lambda_{ij}, \tag{2}$$

with hazard rate

$$\lambda_{ij} = \exp\left(\beta \cdot X_{ij}\right)\lambda_0, \tag{3}$$

where $\lambda_0$ is the baseline hazard rate and $\beta$ is the effect of time in close proximity or any other covariate $X_{ij}$ on the log rate ratio. If $\beta > 0$ an increase in the covariate would accelerate (increase the risk or rate of) transmission, while if $\beta < 0$ an increase in the covariate would decelerate (decrease the risk or rate of) transmission. The cumulative risk of transmission further increases with prolonged exposure (higher $\tau_{ij}$).

The baseline hazard rate consists of two terms

$$\lambda_0^{1-I_i=0} \mu_0^{I_i=0} \tag{4}$$

where $\lambda_0^{1-I_i=0}$ is the rate for internal transmissions ($i > 0$ denoting an internal source) and $\mu_0^{I_i=0}$ is the rate for external transmissions ($i = 0$ for an external source). The internal and external rates were modelled with exponential distributions. The values of covariates not defined for external pairs (close proximity, shared classroom time, and air quality) were set to zero, and the effects of covariates defined also for external pairs (social factors) included an interaction term, so that these covariates could have different effects on the rate of internal and external transmissions.

The pairwise accelerated failure time regression model was estimated under the assumption that who-infected-whom was not observed[54]. Note that a susceptible student could have multiple possible infectors per dataset. The pairwise model considers that by integrating the log likelihoods over all possible infectors i of the susceptible student[54–56].

### Infectiousness-weighted time in close-proximity

We estimated the association between transmission and the daily average time in close proximity during the exposure period weighted by relative infectiousness of the infectious student. Thereby, we aimed to give higher weight to close-proximity times on days when the infectious student was probably more infectious.

We assumed that infectiousness peaked at the day of symptom onset and quickly decreased thereafter. We modelled relative infectiousness using a Weibull distribution with shape parameter $k = 2$ and scale parameter $\lambda = \frac{t_p}{\left(\frac{k-1}{k}\right)^{\frac{1}{k}}}$ depending on the day of symptom onset $t_p$ (Supplementary Fig. S17). The probability weights $p_d$ for days $d = 1, \ldots, T_i$ of the infectious period of individual $i$ were sampled from a Weibull(2, $\lambda | t_p$). They were discretised as $\tilde{p} = \int_0^{1.5} p(t)dt$ for $d = 1$ and $\tilde{p} = \int_{d-0.5}^{d+0.5} p(t)dt$ for $d > 1$, and divided by $p_T = \int_0^{T_i+0.5} p(t)dt$ so that $\sum_d p_d = \sum_d \frac{\tilde{p}_d}{p_T} = 1$. The scaling factors $w_d$ were obtained by multiplying $p_d$ with $T_i$. Each daily close contact time $X_{ij}^d$ during the infectious period was up- or down-weighted by multiplying with $w_d$ and the sum of the re-scaled close-proximity times $Y_{ij}$ was computed. $Y_{ij}$ was then divided by the length of the exposure period $T_j$ to obtain the daily average infectiousness-weighted close-proximity time $\bar{Y}_{ij}$. Formally,

$$p_d \sim \text{Weibull}\left(2, \lambda | t_p\right) \tag{5}$$

$$w_d = p_d \cdot T_i \tag{6}$$

$$Y_{ij} = \sum_{d=1}^{T_i} w_d \cdot X_{ij}^d \tag{7}$$

$$\bar{Y}_{ij} = \frac{1}{T_j} \cdot Y_{ij} \tag{8}$$

We show the impact of weighting by infectiousness with examples in Table S2.

Finally, we computed the log of the daily average infectiousness-weighted time in close proximity. The log reduced the skew in the distribution of close-proximity times and the impact of outliers from rare but very long close-range interactions. By using the log, the estimated effect remains interpretable, giving the rate ratio for a doubling of the time in close proximity. Note that other time-varying risk factors

(time in the same classroom and suboptimal air quality) were also weighted by infectiousness and the log transformation applied.

## Model estimation

We generated 1000 paired datasets using a probabilistic approach considering uncertainty in the timing of infections and infectiousness. We also estimated the contribution of risk factors by comparing the relative changes in the model log-likelihoods when adding each factor to the null model without covariates. Pairwise accelerated failure time models were estimated using the *TranStat* package version 0.3.7[63]. All analyses were performed in R version 4.2. We report continuous variables with median and interquartile range (IQR), categorical variables with frequency and proportion (%), and model-based estimates with the median of the means and 95%-confidence intervals (CIs) across datasets.

## Ethics statement

The study was approved by the Ethics Committee of the Canton of Bern, Switzerland (reference no. 2023-02035). All students willing to participate in the collection of molecular, epidemiological, and proximity data were included, and written informed consent was obtained from the students and their caregivers.

## Reporting summary

Further information on research design is available in the Nature Portfolio Reporting Summary linked to this article.

## Data availability

De-identified close-proximity data and all genomic sequences generated from the data collected during this study are available at osf.io/naut4. Sequences used in the main analysis detailed in this manuscript are further available on GenBank under accession numbers detailed in Supplementary Text C. Genomic viral sequences collected as part of the Swiss Respiratory Virus Sequencing study are available in BioProject PRJEB83635 under the accession numbers detailed in Supplementary Text C and Tables S3-S4. Global genomic viral sequences are available from GenBank (https://www.ncbi.nlm.nih.gov/genbank/) under the accession numbers detailed in Supplementary Text C and Tables S5-S6. Restrictions on the availability of other personal data apply but are necessary to maintain the confidentiality of participants. The data can be made available upon request, subject to approval by the local ethics committee and the Technology Transfer Office of the University of Bern. Requests will be given expedited review, but the timeframe depends on the review time of the involved parties (contact: University of Bern, info.ispm@unibe.ch).

## Code availability

The R code files for the descriptive analysis, generation of the paired datasets used for modelling, and the analysis of the modelling results are available at: osf.io/naut4. Via the same link, we also provide code files and a dummy dataset to run the main functions used to generate and analyse the paired dataset. This enables users to test our data processing and analysis pipeline, as it is not possible to reproduce the main modelling results due to restrictions on the availability of personal data. All analyses are performed in *R* version 4.2. Pairwise accelerated failure time models are estimated using the *TranStat* package version 0.3.7. To generate and analyse the paired dummy datasets, the following additional R packages are used: *tidyverse* version 2.0.0, *tidygraph* version 1.3.1, *ggraph* version 2.2.1, and *lubridate* version 1.9.4. All required R packages are automatically installed when running the code files.

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

## Acknowledgements
This study was funded by the Multidisciplinary Center for Infectious Diseases (MCID), University of Bern, Bern, Switzerland. N.B. was supported by funding from the Swiss National Science Foundation (grant no. P500PM_230473). T.S., J.D.M., and C.B. acknowledge funding from ETH Zürich and the Swiss National Science foundation (CRSII5_205933). C.C. and L.D. acknowledge partial funding from the Lagrange Project of ISI Foundation funded by CRT Foundation and from the EPFL COVID19 Real Time Epidemiology I-DAIR Pathfinder funded by Fondation Botnar. We would further like to thank the school, teachers, and students who participated in the study. We are also grateful to the student assistants who helped with the data collection in the schools. We would like to acknowledge the support of the Functional Genomics Center at ETH Zurich for the genomic sequencing of our saliva samples.

## Author contributions
Conception and design: N.B., L.F., P.B., P.J., T.H., J.M. Data collection: N.B., L.F., P.J., T.H., C.C., L.D. Laboratory analysis: P.B., L.Fu. Genomic analysis: J.M., C.B., TS. Contact network analysis: C.C., L.D., N.B., J.M. Statistical analysis and modelling: N.B., J.M. First manuscript draft: N.B., J.M., L.F., M.E., T.S. All authors reviewed and approved the final version of the manuscript.

## Competing interests
The authors declare no competing interests.
