## [Peer review File · Nature Communications]

The relative contribution of close-proximity contacts, shared classroom exposure and indoor air quality to respiratory virus transmission in schools

Corresponding Author: Professor Lukas Fenner

Version 0:

Reviewer comments:

Reviewer #1

(Remarks to the Author)

This is a state of the art epidemiological study combining several proven data collection and analysis techniques to estimate human-to-human transmission risk of respiratory pathogens.

While results from several of these techniques have been reported on in the past, their combination adds to the literature in an important way.

I applaud the authors for its successful completion and for their comprehensive analysis and balanced discussion.

(Remarks on code availability)

Reviewer #2

(Remarks to the Author)

In this paper, Banholzer et al. provide new evidence about the risk factors for respiratory virus transmission in classroom settings. By studying a small cohort (67 students) over a 5-6 week period in a Swiss secondary school and inferring transmissions between the students, they are able to provide some estimates about the relative risks for virus transmission of close proximity, time spent together indoors, and air quality. The study is detailed and well executed, with its limitations clearly stated. In light of some of these limitations it is unclear to me how easily the results themselves translate to other settings, but the method, which carefully combines molecular, epidemiological, proximity and environmental data, is certainly a valuable contribution which has potential to be applied to many other studies in different settings. This is particularly true because the code has been clearly documented and shared. I recommend the paper for publication.

Some comments:

Lines 42-55: the outline of the existing literature in the first two paragraphs could benefit from a little more clarity: are all the claims substantiated for both SARS-CoV-2 and Influenza? Is one better studied than the other, for any of these aspects? How about the other respiratory viruses which occurred in your study, is anything known about these? One of the citations (17) appears to be about multidrug-resistant Mycobacterium tuberculosis, which has not been mentioned in the text so far.

Why was 1.5m picked as the threshold for “close-proximity”? The sensitivity analysis in the supplementary material shows it to be a reasonable choice, I am wondering whether it was decided a priori, and if so is there a reference for using this distance?

Could the authors comment a little more on the school setup? I am interested in how “between class” transmissions could have occurred and wonder if more detail could be added about when and how the classes would have met – would it be only outside? In a big hall? Passing in corridors?

In the overall findings and much of the discussion, various respiratory viruses are considered together; until lines 214-218 there seems to be an implicit assumption that the risk factors apply similarly across the pathogens under consideration. Can

any of the results be split out by pathogen or do they all present small sample size problems?

Lines 240-242 are, I think, key for the interpretation of the results. I suggest they could be brought forward in the manuscript, e.g. in the abstract's final line you could consider something like "Prolonged exposure in shared, poorly ventilated spaces, which potentially includes several infectious sources, drives respiratory virus transmission more than close contact with a single source."

Do you have data on exactly when the close proximity contacts happened? In particular, were they during the times in shared classrooms, and do you know what the air quality was like at those exact times? It would be great if it were possible to add something about the interaction between these risk factors.

(Remarks on code availability)

I have viewed the code and it appears to be clear and well documented. There is a wiki file which explains how to use the repository. I have not tried installing and running the code; as far as I can see, this would require me having access to the data (which is available on request). It would be nice if the authors could provide a dummy data set, sufficient for the code to run; this helps future users who do not need access to the original data but who would like to set up their own analysis using a similar format of csv file etc.

Reviewer #3

(Remarks to the Author)

Review of "The relative contribution of close-proximity contacts, shared classroom exposure and indoor air quality to respiratory virus transmission in schools" by Banholzer et al

Summary

This manuscript is about an analysis of dataset on respiratory virus transmission in a Swiss school, in which the authors estimate the contribution of various contact and environmental factors to transmissibility. Specifically, they try to establish the role close-contact transmission vs environmental transmission in closed spaces. The dataset is rich in that it contains much information on individual level, so that a reliable reconstruction can be made of what has happened in terms of infections and the underlying contributing factors. The analysis is done in two steps: first, epidemiological, sample, and sequence data are used to create 1,000 datasets with exact infection times and infectiousness distributions, which are translated to datasets of potential transmission events (paired datasets); second, the paired datasets are used in a survival model to estimate the association of transmission with explanatory variables from air quality and proximity data. The authors conclude that the role of close proximity contacts to transmission is limited, and that sharing classrooms, especially if not so well ventilated, poses a larger risk.

I have enjoyed reading the paper and really like the approach. I think the authors make clever use of all information: saliva to indicate transmission events, epidemiological and sequence information to exclude transmission events and times, and a survival analysis to include all pairwise exposures with associated factors into single outcomes of infection (or not).

General comment

From what I understand, is that in each sampled dataset, external transmission pairs were only defined if a case was not internally linked. I would expect that to lead to an underestimation of external transmission, because it forces a case to have been infected internally if there was any infector present at the time of infection. Especially with RSV, of which the outbreak was already ongoing at the start of the study, can't that lead to incorrect infectors. Especially if close contact plays an important role and the actual infection event was before the start of the study, the relevant close contacts may have been missed and transmissions are now ascribed to shares space. Please reflect on this in the discussion (or run analyses allowing external infection also if cases are internally linked).

Specific comments.

Line 100: "15 plausible transmission pairs for IAV and 9 for RSV". First, refer to S3 and S6 (not S5). But then, in these figures I count 12 and 6 pairs (number of dark "TRUE" boxes on one side of the diagonal)

line 318: I think $l(s_1, s_2)$ and $d(s_1, s_2)$ are not defined in the text

line 439: who-infected-whom

Figure 1: from the figure it seems as if all "collected data" are used to make "paired input data", which are then used in the "pairwise survival model". But that is not the case. Only the molecular+epidemiological data are used in step 2; the other data are used in step 3. Can you make that clear, e.g. by separating the two groups of datasets (e.g. dotted line), place the arrow (1) a bit higher, and add a second arrow (2) from the bottom two datasets directly to the right.

Also, at the bottom right, $\beta * X_{ij}$ is not only the proximity effect but also the air quality effect (and other covariates), isn't it? And, the external rate should have $l_{i=0}$ in the superscript.

Figure 2, caption: should end with "See figure 5...", not 3

(Remarks on code availability)

I have looked at the code, seen that it is available and that it seems well-written. I have not tried to run anything, which may even not be possible because not all data are open.

For all data that are (will be available), is it possible to download the code + data in an R-project format so that relative paths to data or sourced functions work properly? It would make reproducibility a lot easier. Please explain.

Response to Reviewer #1

Comment 1.0: This is a state of the art epidemiological study combining several proven data collection and analysis techniques to estimate human-to-human transmission risk of respiratory pathogens.

While results from several of these techniques have been reported on in the past, their combination adds to the literature in an important way.

I applaud the authors for its successful completion and for their comprehensive analysis and balanced discussion.

Response 1.0: Thank you very much for your positive assessment of our work.

Response to Reviewer #2

Comment 2.0: In this paper, Banholzer et al. provide new evidence about the risk factors for respiratory virus transmission in classroom settings. By studying a small cohort (67 students) over a 5-6 week period in a Swiss secondary school and inferring transmissions between the students, they are able to provide some estimates about the relative risks for virus transmission of close proximity, time spent together indoors, and air quality. The study is detailed and well executed, with its limitations clearly stated. In light of some of these limitations it is unclear to me how easily the results themselves translate to other settings, but the method, which carefully combines molecular, epidemiological, proximity and environmental data, is certainly a valuable contribution which has potential to be applied to many other studies in different settings. This is particularly true because the code has been clearly documented and shared. I recommend the paper for publication.

Response 2.0: Thank you very much for your great summary and recommendation.

Comment 2.1: Some comments:

Lines 42-55: the outline of the existing literature in the first two paragraphs could benefit from a little more clarity: are all the claims substantiated for both SARS-CoV-2 and Influenza? Is one better studied than the other, for any of these aspects? How about the other respiratory viruses which occurred in your study, is anything known about these? One of the citations (17) appears to be about multidrug-resistant Mycobacterium tuberculosis, which has not been mentioned in the text so far.

Response 2.1: Thank you, we have used this opportunity to provide more details about the existing literature. The paragraph in the revised Introduction now reads as follows (page 6):

“Several studies have assessed close-contact patterns in schools¹⁰⁻¹², yet without quantifying their effect on transmission, especially not in relation to other risk factors. Empirical studies are scarce but indicate that close-range interactions may be needed for transmission¹³⁻¹⁸, although prolonged exposure at longer ranges may incur a similar risk¹³. Specifically, proximity was associated with infection with SARS-CoV-2 during a long-distance train ride¹⁶ and Mycobacterium tuberculosis during an airplane flight¹⁷. While most studies relate to SARS-CoV-2^{13,14,16,18}, recent work used a viral challenge model to study the transmission of multiple respiratory viruses (influenza virus A virus, respiratory syncytial virus, adenovirus, etc) following 30min close-contact interactions between infectious children and susceptible adults¹⁵. Evidence from the SARS-CoV-2 pandemic further suggests that indoor air quality plays a critical role in the transmission of respiratory infections¹⁹. However, the relative importance of different risk factors remains inadequately quantified, limiting our ability to design targeted interventions for effective public health recommendations.”

Comment 2.2: Why was 1.5m picked as the threshold for “close-proximity”? The sensitivity analysis in the supplementary material shows it to be a reasonable choice, I am wondering whether it was decided a priori, and if so is there a reference for using this distance?

Response 2.2: The wearable sensors detect close proximity over varying distances. During post-processing, different attenuation thresholds can be chosen, each corresponding to an approximate distance of close proximity. We chose an attenuation threshold of -65 dBm as the default, corresponding to a proximity of approximately 1.5m, given the radio setup and enclosures used in our data collection. The choice was made before the analyses began, motivated by previous studies in similar settings and because the threshold is close to the resolution limit of the sensing technology. However, to ensure our results are robust to small variations in the attenuation threshold, we also conducted a sensitivity analysis with a slightly higher and lower one. We explain the rationale for our choice in the revised Methods (page 15):

“For the present study, data were post-processed with an attenuation threshold of -65 dBm so that close-range interactions within about 1-1.5 metres were detected. This choice was made before analysis began, considering the radio setup and enclosures used for data collection, and following the choices made in previous studies with similar setups and settings^{41,48}. To assess variation, we performed a sensitivity analysis with a slightly higher and lower threshold compared to our default, which detects more and less close-proximity encounters at larger and shorter distances, respectively.”

Comment 2.3: Could the authors comment a little more on the school setup? I am interested in how “between class” transmissions could have occurred and wonder if more detail could be added about when and how the classes would have met – would it be only outside? In a big hall? Passing in corridors?

Response 2.3: We agree that more details about the school setup may be needed as it can vary between countries and age groups. In response to your comment, we have included a schematic figure and provided more explanation (see new Figure 7 copied below and Methods, page 14):

“The classes were located on the same building floor (Figure 7), allowing students of different classes to interact during breaks and personal study. On a typical weekday, the students arrived at the school around 7:30am and lessons ran until noon, with small 5-10min breaks between lessons and a longer 30min break at 10am, during which they wore the sensors. After a one-hour lunch break, often outside school and without wearing the sensors, lessons continued in the afternoon until about 3:30pm. Most indoor lessons took place in shared classrooms among students of the same class, but students from different classes could interact during short breaks in the corridor, long breaks on the school ground, and during times reserved for personal study in shared rooms.”

Figure 7: Schematic view of the school study setup. The four classrooms were on the same floor of the building, separated by a corridor. There were also rooms for personal study and group work, where students from different classes could meet. During breaks, students could choose to spend their time indoors or outside on the schoolground. Each room has a door onto the corridor and a window overlooking the surrounding schoolground.

Comment 2.4: In the overall findings and much of the discussion, various respiratory viruses are considered together; until lines 214-218 there seems to be an implicit assumption that the risk factors apply similarly across the pathogens under consideration. Can any of the results be split out by pathogen or do they all present small sample size problems?

Response 2.4: It would be interesting to explore variation between pathogens. Unfortunately, our sample size per pathogen does not permit estimation of pathogen-specific effects. We acknowledge this as a limitation in our revised Discussion (page 13):

“Third, although more participants across multiple schools would strengthen our results, our data-intensive longitudinal design with high-frequency sampling provides an in-depth analysis that compensates for breadth. We note, however, that the sample size per pathogen was not large enough to study variation in the effects of transmission risk factors between pathogens.”

Comment 2.5: Lines 240-242 are, I think, key for the interpretation of the results. I suggest they could be brought forward in the manuscript, e.g. in the abstract’s final line you could consider something like “Prolonged exposure in shared, poorly ventilated spaces, which potentially

includes several infectious sources, drives respiratory virus transmission more than close contact with a single source.”

Response 2.5: Thank you, we have included this caveat in the final line of the revised abstract.

Comment 2.6: Do you have data on exactly when the close proximity contacts happened? In particular, were they during the times in shared classrooms, and do you know what the air quality was like at those exact times? It would be great if it were possible to add something about the interaction between these risk factors.

Response 2.6: This is an intriguing question, thank you. We know from our data when exactly close-proximity encounters occurred and how the air quality was during these times. We expect that most contacts occurred in shared classrooms because the students spent most of their time with their classmates during indoor lessons (see also Response 2.3). We agree that exploring interactions between risk factors would be interesting, but it is not immediately clear to us how to study them or whether the paired survival analysis framework would be the most suitable approach. We believe that interactions between risk factors is just one of many possible additional analyses of this extensive, individual-level and multimodal dataset we have collected. We thank the Reviewer for this thoughtful suggestion and will consider it, along with other analyses, for future work.

Comment 2.7: I have viewed the code and it appears to be clear and well documented. There is a wiki file which explains how to use the repository. I have not tried installing and running the code; as far as I can see, this would require me having access to the data (which is available on request). It would be nice if the authors could provide a dummy data set, sufficient for the code to run; this helps future users who do not need access to the original data but who would like to set up their own analysis using a similar format of csv file etc.

Response 2.7: Thank you, this is a great idea. We have provided such a dataset along with code to generate and analyse it. In the revised manuscript, we refer to it in the ‘Code availability’ section (page 22):

“The R code files for the descriptive analysis, generation of the paired datasets used for modelling, and the analysis of the modelling results are available at: osf.io/naut4. Via the same link, we also provide code files and a dummy dataset to run the main functions used to generate and analyse the paired dataset. This enables users to test our data processing and analysis pipeline, as it is not possible to reproduce the main modelling results due to restrictions on the availability of personal data.”

Response to Reviewer #3

Comment 3.0: Review of “The relative contribution of close-proximity contacts, shared classroom exposure and indoor air quality to respiratory virus transmission in schools” by Banholzer et al

Summary

This manuscript is about an analysis of dataset on respiratory virus transmission in a Swiss school, in which the authors estimate the contribution of various contact and environmental factors to transmissibility. Specifically, they try to establish the role close-contact transmission vs environmental transmission in closed spaces. The dataset is rich in that it contains much information on individual level, so that a reliable reconstruction can be made of what has happened in terms of infections and the underlying contributing factors. The analysis is done in two steps: first, epidemiological, sample, and sequence data are used to create 1,000 datasets with exact infection times and infectiousness distributions, which are translated to datasets of potential transmission events (paired datasets); second, the paired datasets are used in a survival model to estimate the association of transmission with explanatory variables from air quality and proximity data. The authors conclude that the role of close proximity contacts to transmission is limited, and that sharing classrooms, especially if not so well ventilated, poses a larger risk.

I have enjoyed reading the paper and really like the approach. I think the authors make clever use of all information: saliva to indicate transmission events, epidemiological and sequence information to exclude transmission events and times, and a survival analysis to include all pairwise exposures with associated factors into single outcomes of infection (or not).

Response 3.0: Thank you for the overall positive assessment of our work.

Comment 3.1: General comment

From what I understand, is that in each sampled dataset, external transmission pairs were only defined if a case was not internally linked. I would expect that to lead to an underestimation of external transmission, because it forces a case to have been infected internally if there was any infector present at the time of infection. Especially with RSV, of which the outbreak was already ongoing at the start of the study, can't that lead to incorrect infectors. Especially if close contact plays an important role and the actual infection event was before the start of the study, the relevant close contacts may have been missed and transmissions are now ascribed to shares space. Please reflect on this in the discussion (or run analyses allowing external infection also if cases are internally linked).

Response 3.1: We acknowledge that external transmissions may be underestimated with our approach and have taken multiple measures to reduce this bias. First, note that internal transmissions are already less likely for RSV because the six index cases at study start cannot be linked internally. Second, we used genomic analysis to exclude transmissions whenever possible (i.e. for IAV and RSV). Third, we generated multiple paired datasets, considering uncertainty in epidemiological parameters, so that there are datasets where two infections are linked internally and others where they are not, depending on the specific parameters. Nevertheless, we agree that

we should put more emphasis on this aspect in our limitations, which we have revised accordingly (page 12–13):

“Second, incomplete genomic sequence coverage for pathogens other than influenza A and respiratory syncytial virus limited the inference of transmission networks. Our probabilistic framework considered a wide range of scenarios, including school absences and uncertainty about the timeline of infections. However, it may still underestimate the number of external transmissions, as any transmission will be categorised as internal if the timeline of the person who became infected (infectee) aligns with that of at least one plausible source of infection (infecter) among the participating students.”

Comment 3.2: Specific comments.

Line 100: “15 plausible transmission pairs for IAV and 9 for RSV”. First, refer to S3 and S6 (not S5). But then, in these figures I count 12 and 6 pairs (number of dark “TRUE” boxes on one side of the diagonal)

Response 3.2: Thank you for bringing our attention to this oversight. This is a typographical error in the manuscript, which we have now amended (page 8):

“[...] we were able to identify 12 plausible transmission pairs for IAV and 6 for RSV (Supplementary Figures S3 and S6).”

Comment 3.3: line 318: I think $l(s_1, s_2)$ and $d(s_1, s_2)$ are not defined in the text

Response 3.3: We thank the reviewer for highlighting this omission. We have now amended the text to detail the definitions in the revised Methods section (page 16):

“To establish likely transmission events, we calculated the number of single nucleotide polymorphisms (SNPs) between each pair of sequences as the proportion of comparable sites. We then computed the likelihood that sequences were consistent with a transmission using the Jukes Cantor ‘69 substitution model. We assumed an evolution time of the number of days between test dates plus a perceived maximum of 4 days (2 days of within-host evolution per sample), and calculated the log-likelihood of the model using the closed form expression ⁵¹

$$\log \mathcal{L}(\mu, t \mid s_i, s_j) = l(s_i, s_j) \log \left(\frac{1}{4} + \frac{3}{4} e^{-\frac{4}{3}\mu t} \right) + d(s_i, s_j) \log \left(\frac{1}{4} - \frac{1}{4} e^{-\frac{4}{3}\mu t} \right),$$

where μ is the substitution rate, t is the time between samples in evolutionary time, s_i and s_j are the sequences of samples collected from individuals i and j , and $d(s_i, s_j)$ and

$l(s_i, s_j)$ are the hamming distance and the number of comparable nucleotide positions between s_i and s_j , respectively. For both IAV and RSV, we assumed a nucleotide substitution rate of 1.5×10^{-3} mutations per site per year, which is close to the estimates in literature^{52,53}. “

Comment 3.4: line 439: who-infected-whom

Response 3.4: We believe the term is common and correctly spelled, thus we are not sure how to resolve this comment. If the reviewer would care to elaborate, we would be happy to reconsider and revise accordingly.

Comment 3.5: Figure 1: from the figure it seems as if all “collected data” are used to make “paired input data”, which are then used in the “pairwise survival model”. But that is not the case. Only the molecular+epidemiological data are used in step 2; the other data are used in step 3. Can you make that clear, e.g. by separating the two groups of datasets (e.g. dotted line), place the arrow (1) a bit higher, and add a second arrow (2) from the bottom two datasets directly to the right.

Also, at the bottom right, $\beta * X_{ij}$ is not only the proximity effect but also the air quality effect (and other covariates), isn't it?

And, the external rate should have $I_i=0$ in the superscript.

Response 3.5: Thank you for these suggestions. We have incorporated them into the revised Figure 1 (copied below with a caption). The figure now distinguishes the survival model more clearly from the paired input data, and indicates the infectiousness weighting step of the time-varying covariates, including close-proximity time and indoor air quality.

Figure 1. Schematic overview of the collected and input data for the pairwise survival model. (1) We detected virus infections in saliva samples and constructed paired datasets with transmissions that were epidemiologically plausible, excluding pairs where either the infectious or exposed student was absent from school. We also excluded transmissions of influenza A and respiratory syncytial virus through genomic analysis. If an infection could not be linked internally to another participating student, it was considered a transmission from an external source such as a household or community member. (2) For each student pair, we determined the time in close proximity during the exposure period, which was collected with wearable sensors worn by students throughout school lessons and break times. For student pairs within classes, we determined time spent in shared classrooms and in suboptimal air quality, which was measured with air quality monitors and aerosol devices. Both covariates were weighted by relative infectiousness of the infector and averaged over the exposure interval, which is the period from the onset of infectiousness of student i until the date of infection of student j. If j is never infected, the exposure period is until the end of the infectious period of i. School-free days and absences are not included in the exposure period. (3) The association of respiratory virus transmission with time in close proximity and other risk factors was estimated using accelerated failure time regression models, with an internal and external rate of transmission.

Comment 3.6: Figure 2, caption: should end with “See figure 5...”, not 3

Response 3.6: Well spotted, thank you.

Comment 3.7: I have looked at the code, seen that it is available and that it seems well-written. I have not tried to run anything, which may even not be possible because not all data are open. For all data that are (will be available), is it possible to download the code + data in an R-project format so that relative paths to data or sourced functions work properly? It would make reproducibility a lot easier. Please explain.

Response 3.7: With the revised manuscript, we have provided a dummy dataset to test the data processing and analysis pipeline (see also our Response 3.7 to a similar comment from Reviewer #2). The ‘test-dummy’ folder in our public data repository includes R code files with relative paths automatically detected from the code file, so that users should be able to run the code directly from the file. The link to the repository is provided in the revised ‘Code availability’ subsection (page 22):

“The R code files for the descriptive analysis, generation of the paired datasets used for modelling, and the analysis of the modelling results are available at: osf.io/naut4. Via the same link, we also provide code files and a dummy dataset to run the main functions used to generate and analyse the paired dataset. This enables users to test our data processing and analysis pipeline, as it is not possible to reproduce the main modelling results due to restrictions on the availability of personal data.”